# Characteristics of Mesoscale Eddies in the Vicinity of the Kuroshio: Statistics from Satellite Altimeter Observations and OFES Model Data

Yiyun Shi [1,2], Xiaohui Liu [2,3,*], Tongya Liu [2,3] and Dake Chen [1,2,3,*]

1   School of Oceanography, Shanghai Jiao Tong University, Shanghai 200240, China
2   State Key Laboratory of Satellite Ocean Environment Dynamics, Second Institute of Oceanography, Ministry of Natural Resources, Hangzhou 310005, China
3   Southern Marine Science and Engineering Guangdong Laboratory (Zhuhai), Zhuhai 519000, China
*   Correspondence: xh_liu@sio.org.cn (X.L.); dchen@sio.org.cn (D.C.)

**Abstract:** Mesoscale eddies propagate westward in the northwestern Pacific Ocean and interact with the Kuroshio in the vicinity of the western boundary of the ocean. However, the processes affecting the eddy properties and the detailed structure of the eddies when they encounter the Kuroshio remain unclear. In this study, we analyze the statistics of the eddy properties around the Kuroshio using 25 years of satellite altimeter data and the eddy-resolving OFES model product. The spatial compositions of the eddies in the northwestern Pacific show that, as the eddies propagate westward, their radius and amplitude decrease sharply when they approach the Kuroshio region. The radius, amplitude, and kinetic energy of the eddies reaching the Kuroshio region decay much faster during their lifespan compared with the eddies in the interior of the Pacific Ocean. Furthermore, the three-dimensional structure of the eddies obtained from the OFES model data shows that the maximum temperature anomalies in the cyclonic and anticyclonic eddies occur at ~300 m, and the maximum depth reduces as a result of the interaction between the eddies and the main Kuroshio current.

**Keywords:** mesoscale eddies; Kuroshio; eddy-mean-flow interaction; satellite altimeter OFES model data

## 1. Introduction

Mesoscale eddies—with typical time scales of tens to hundreds of days and spatial scales of tens to hundreds of kilometers—are dominant features in the upper ocean [1,2]. Because the eddies carry enormous kinetic energy, which is usually an order of magnitude larger than that of the mean flow [3,4], and propagate westward in the upper ocean, the characteristics of the eddies during their lifecycle are of great importance for energy transport and transformation. Liu et al. [5] analyzed mesoscale eddies in the northern Pacific Ocean and divided an eddy's lifespan into young, mature, and aged stages. Chelton et al. [6,7] investigated the statistical features and propagation characteristics of mesoscale eddies using satellite altimeter data. They suggested that the average amplitude of the eddies is 8 cm and the average radius is 90 km. They also found that anticyclonic eddies live longer and propagate farther than cyclonic eddies.

Mesoscale eddies formed in the open ocean propagate westward under the beta effect [8–10]. In the vicinity of the western boundary of an ocean basin, the eddies interact with the western boundary current of the basin. Zhai et al. [11] suggested that most westward-propagating mesoscale eddies do not escape from the western boundary region, and thus, they called the western boundary region the "graveyard" of the eddies. They also revealed that the kinetic energy carried by the eddies is transferred to higher-wavenumber vertical modes and is then dissipated. Therefore, the characteristics and variability of mesoscale eddies in the vicinity of the western boundary are worth investigating.

The Kuroshio is a western boundary current of the northwestern Pacific Ocean. Eddies in the northwestern Pacific Ocean propagate westward, and some reach the Kuroshio region [12]. The eddies and the Kuroshio interact near the western boundary of the Pacific Ocean [13–15]. Using a reduced-gravity primitive equation, Kuo and Chern [16] found that a cyclonic eddy loses its energy to the mean field, whereas an anticyclonic eddy can obtain energy from the mean flow during interaction with the western boundary current. However, by combining satellite altimeter data and model outputs, Yan et al. [17] suggested that both cyclonic and anticyclonic eddies are deformed into an elliptic shape with the major axis in the northeast–southwest direction and obtain kinetic energy from the mean flow in the interaction. Using a fine-resolution regional general circulation model, Shi et al. [18] discussed the influences of the eddies on the main current of the Kuroshio.

In this study, we analyze the characteristics of the mesoscale eddies in the vicinity of the Kuroshio region and compare them with the eddies in the Pacific interior. We find that the decay processes of the eddies in the Kuroshio region, evaluated by the amplitude, size, and vorticity of the eddies, differ markedly from those in the Pacific interior. We also depict the evolution of the 3D structure of the eddies as they approach the Kuroshio. The remainder of the paper is organized as follows: Section 2 describes the data and the eddy detection scheme used in this paper; Section 3 presents a statistical analysis of the eddy dataset; Section 4 discusses the different influences of the Kuroshio on cyclonic and anticyclonic eddies, and a summary is presented in Section 5.

## 2. Data and Methods

### 2.1. Satellite Altimeter Data

The sea surface height (SSH) data used for eddy detection and statistical analysis in this study are the satellite altimeter data produced by the Archiving, Validation, and Interpretation of Satellite Oceanographic data (AVISO). The data can be downloaded from http://marine.copernicus.eu (accessed on 28 May 2021). The dataset contain the SSH and surface geostrophic current fields, with a horizontal resolution of $1/4° \times 1/4°$ as daily data from 1993 to 2017. The surface geostrophic velocity of the AVISO is derived from the SSH data by the geostrophic balance as follows:

$$u = -\frac{g}{f}\frac{\partial h}{\partial y}, \; v = -\frac{g}{f}\frac{\partial h}{\partial x}$$

where $u$ and $v$ are the surface geostrophic velocity components in the $x$- and $y$-directions, $h$ is the sea surface height (SSH), $f$ is the Coriolis parameter, $g$ is the gravitational acceleration, and $x$ and $y$ are the zonal and meridional directions, respectively.

### 2.2. Three-Dimensional Eddy-Resolving Model Data

The Ocean general circulation model for the Earth Simulator (OFES) data are used to analyze the 3D structure of the eddies [19]. The data can be downloaded from http://apdrc.soest.hawaii.edu (accessed on 15 August 2021). The variables used in this study include the SSH, ocean temperature, and 3D velocity field. The horizontal resolution of the OFES data is $0.1° \times 0.1°$, and the temporal interval is 3 days; the dataset covers the period from 1993 to 2017. The detailed model setup and validations of the dataset may be found in [19].

### 2.3. Eddy Detection Scheme

The eddy detection scheme used in this study is based on the velocity vector geometry method [20] and was proposed by Dong et al. [21]. Compared with other eddy detection methods (i.e., the winding-angle method [1,7,22] or the Okubo–Weiss method [23–25]), the vector geometry method guarantees a higher accuracy and lower false detection rate and is more flexible and easy to use in any velocity field [20]. It has been applied to eddy detection in the wakes of the Hawaiian Islands [21] and in the South China Sea [26]. In brief, the eddy detection scheme can be divided into three steps: first, the eddy center

location is obtained using four constraints that guarantee the velocity sign changes along the east–west (north–south) direction with the minimum velocity in the vicinity of the center; second, the boundary of the eddy is obtained by calculating the stream function contour and choosing the outermost closed contour around the center; lastly, the method ensures that the same eddy is tracked by requiring the same polarity (cyclonic or anticyclonic) in a suitably sized area that is not so small that it divides the same eddy trajectory into multiple eddy trajectories, but not so large that other eddies are processed into the same eddy. Further details of the tracking method have been described by Doglioli et al. [27] and Chaigneau et al. [1]. The radius, amplitude, time, EKE, vorticity, central location, and boundary are obtained for detected eddies.

The study area extends from 120° E in the west to 135° E in the east and from 15° N to 27° N (Figure 1). This area is in the core of the Subtropical Countercurrent (STCC) [28] and is rich in eddies with a higher EKE (Figure 1b). There is strong energy exchange between the westwardly propagating eddies and the Kuroshio [5,29,30]. The EKE is given by

$$\text{EKE} = \frac{1}{2}\rho_0\left(u'^2 + v'^2\right)$$

where $u'$ and $v'$ are the surface geostrophic velocity components' disturbance in the *x*- and *y*-directions by subtracting the time average and $\rho_0 = 1025$ kg m$^{-3}$ is the constant reference density.

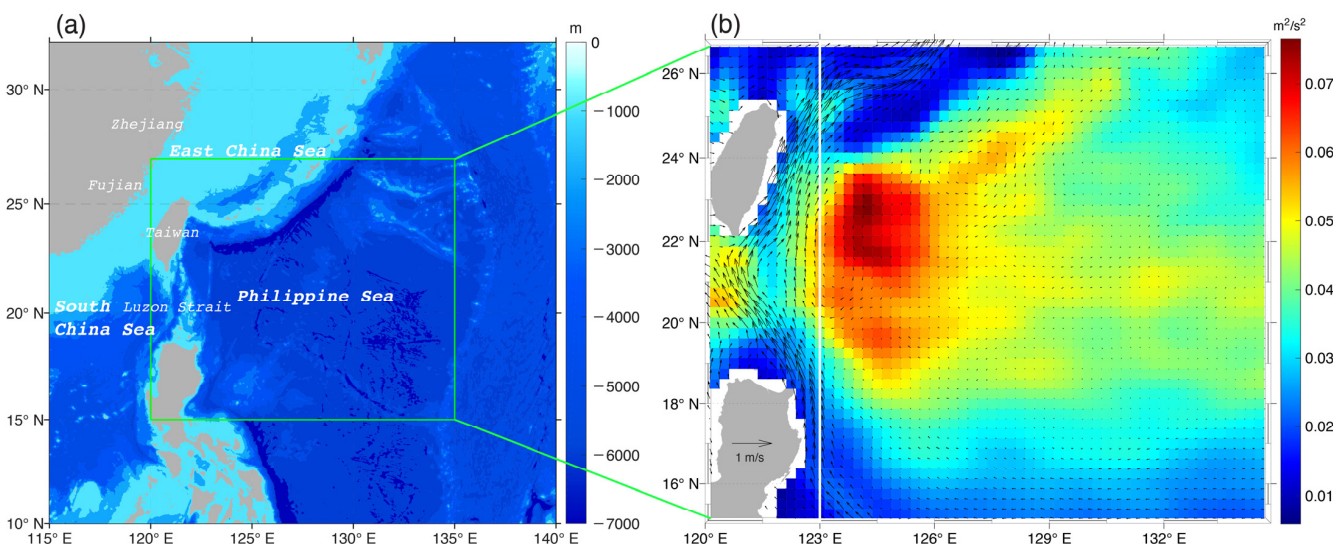

**Figure 1.** (**a**) Bathymetry (m) of the northwest Pacific Ocean. (**b**) Mean surface geostrophic current (black arrows) and eddy kinetic energy (EKE, m$^2$/s$^2$) in the vicinity of the Kuroshio from the 25-year AVISO altimeter data. The white line represents the boundary of the regions where the eddies interact with the Kuroshio or not.

The mean width of the Kuroshio is approximately 100 km [31], and the mean radius of the mesoscale eddies in the global ocean is about 90 km, although the maximum radius can exceed 150 km [7]. The eddies detected in this region are divided into two groups, those that interact with the Kuroshio and those in the Pacific interior. We assume that eddies begin to interact with the Kuroshio at 125° E (dashed line in Figure 2); thus, the track of an eddy that interacts with the Kuroshio is located in the region west of 125° E, and the termination point of the eddy is west of 123° E. The selected trajectories of the cyclonic and anticyclonic eddies that can reach the Kuroshio are presented in Figure 2. In total, there are 125 eddies, of which 69 are cyclonic and 56 anticyclonic. Eddies generated around the main current of the Kuroshio were not considered.

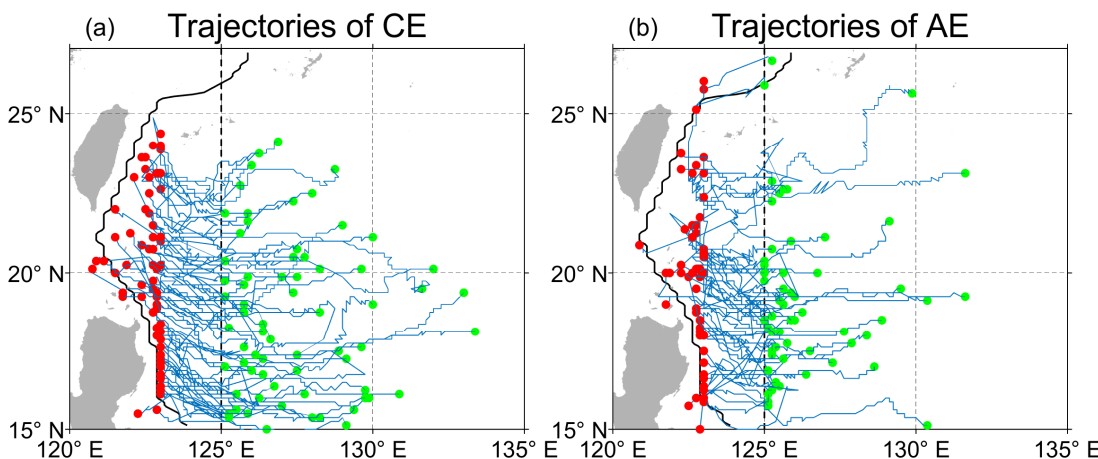

**Figure 2.** Trajectories of the detected (**a**) cyclonic and (**b**) anticyclonic eddies. The solid black line represents the maximum velocity axis of the surface of the Kuroshio, with the velocity fields obtained by averaging the AVISO sea surface geostrophic velocity from 1993 to 2017. The dashed black line is the starting position of the eddy–the Kuroshio interaction. Green (red) dots represent the locations of eddy generation (termination).

## 3. Eddy Analysis in the Kuroshio Region

### 3.1. Eddy Size and Lifespan

The total number of eddies detected in our study area with lifespans longer than 7 days is 26,850 between 1993 and 2017, including 13,342 cyclonic eddies and 13,508 anticyclonic eddies. The difference of the numbers between cyclonic eddies and anticyclonic eddies is insignificant, i.e., only 1.2% more anticyclonic eddies than cyclonic eddies. The number of eddies decreases with increasing lifespan (Figure 3a). The difference of the characteristics between cyclonic eddies and anticyclonic eddies are also insignificant. The average lifespan of both the cyclonic eddies and anticyclonic eddies is about 23–24 days. Eddies with lifespans shorter than 25 days account for about 70% of both cyclonic and anticyclonic eddies, whereas eddies with lifespans longer than 50 days account for nearly 9–10% of all the eddies.

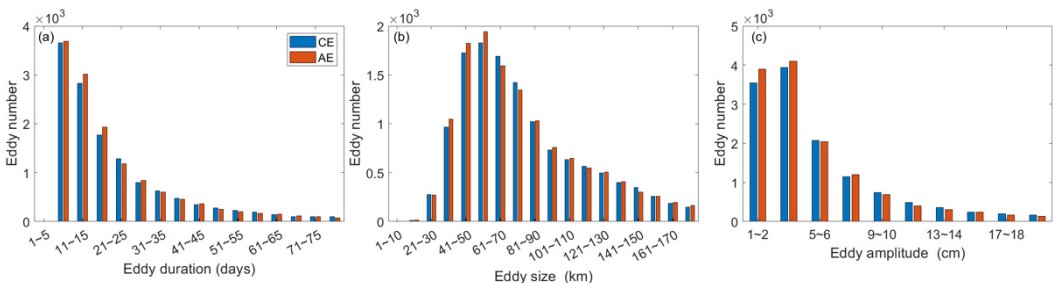

**Figure 3.** Histogram of the (**a**) lifespan, (**b**) maximum size, and (**c**) maximum amplitude for cyclonic and anticyclonic eddies in their lifespan.

The histogram of the eddy numbers with respect to the maximum radius (Figure 3b) peaks at 61–70 km for both cyclonic and anticyclonic eddies, with the total number being about 2000. Eddies with a maximum radius of <70 km during their lifespan make up 51–52% of the eddies. Eddies larger than 100 km account for only ~24% of both cyclonic and anticyclonic eddies, which is consistent with the eddy lifespan distribution. The average eddy sizes for cyclonic and anticyclonic eddies are about 84–85 km, with no significant difference between the two types of eddies. The maximum amplitude in the lifespan of each eddy is smaller than 10 cm for 85.80% of cyclonic eddies and 88.29% of anticyclonic eddies. In contrast, only 3.47% (2.59%) of cyclonic (anticyclonic) eddies have

an eddy amplitude greater than 20 cm. The average amplitude of the cyclonic (anticyclonic) eddies is 5.53 cm (5.19 cm).

### 3.2. Eddy Spatial Distribution

Because the eddies propagate westward, the number of eddies increases toward the western boundary of the Pacific (Figure 4a,e). In the vicinity of the Kuroshio region east of the Luzon Strait and Taiwan, eddies are most frequently observed in the area between the south of Taiwan and north of Luzon Island. There is also a high eddy occurrence off northeastern Taiwan (25°–26° N, 122°–124° E). There is a band-like northeast–southwest zone on the offshore side of the Ryukyu islands, which suggests blocking of the eddies by the island chain. Near the main current of the Kuroshio, the number of eddies is much smaller.

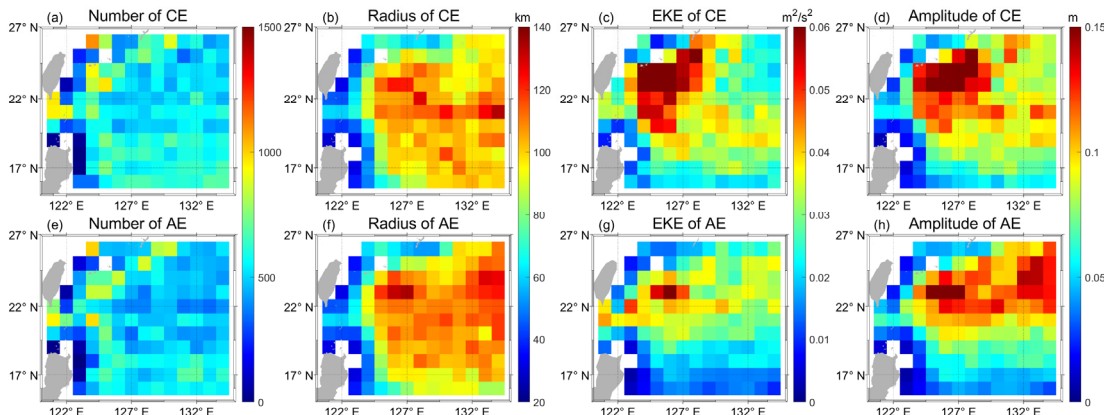

**Figure 4.** Distributions of eddy number (**a**,**e**), radius (**b**,**f**), the EKE (**c**,**g**), and amplitude (**d**,**h**). Values in each panel are averaged within $1° \times 1°$ grid boxes where the eddy tracks snapshot are located. The boxes whose average depth is less than 300 m are masked to avoid unrealistic values. Upper panels show the distributions of cyclonic eddies (CEs), whereas lower panels show those of anticyclonic eddies (AEs).

The eddy size clearly decreases in the Kuroshio region, which suggests that the eddies decay quickly as they interact with the Kuroshio current. Anticyclonic eddies have a larger radius than cyclonic eddies (Figure 4b,f), and the amplitude is smaller around the Kuroshio (Figure 4d,h), which is consistent with Brokaw et al. [32], who found that anticyclonic eddies were generally characterized by a smaller amplitude and larger radius than their cyclonic counterparts.

The spatial distribution of the EKE shows that there is a maximum east of Taiwan for both cyclonic and anticyclonic eddies away from the Kuroshio region, and the EKE of cyclonic eddies is higher and more concentrated than that of anticyclonic eddies. Liu et al. [5] suggested that the high EKE around the east of Taiwan is caused by the abundant background EKE, which is generated through frontal instability associated with the subtropical front.

The distribution of the eddy amplitude also exhibits a maximum east of Taiwan away from the Kuroshio region. The spatial distribution of eddy amplitudes demonstrates that large-amplitude eddies occur preferentially in the western Pacific interior.

The spatial distribution of eddy characteristics near the Kuroshio suggests that the eddies are affected by the Kuroshio as they propagate westward and approach the main current of the Kuroshio. To show the difference in the eddy characteristics between the Kuroshio region and the ocean interior, we further analyzed the zonal variation of the eddies near the Kuroshio region by averaging the values over several meridional bands in the study area, with the width of each band being 1° longitude (Figure 5). Both eddy size and amplitude remain stable from 135° E to 125° E, then decrease quickly from 125° E

to the Kuroshio for both anticyclonic eddies and cyclonic eddies. They reach their lowest values at 121°–122° E. It is worth noting that the radius and amplitude of both cyclonic and anticyclonic eddies increase slightly near 120°–121° E. Previous studies have suggested that the eddies east of the Luzon Strait could affect the looping path of the Kuroshio and generate eddies on the west part of the strait [33,34]. The rise of the eddy radius and strength may be induced by the generation of the eddies in the strait. The variations of the eddy size and amplitude near the Kuroshio suggest that the eddies tend to decay under their interaction with the main current of the Kuroshio. Furthermore, as the eddies propagate westward, the EKE of the eddies increases around the Kuroshio because both cyclonic and anticyclonic eddies can obtain kinetic energy from the Kuroshio during the interaction [17]. Similar to the radius and amplitude in the Luzon Strait from 121° E to 120° E, the EKE also increases slightly, which is also the result of the eddy activity and/or the variation of the looping path of the Kuroshio in the west part of the strait. It is worth noting that the variation of the EKE between cyclonic eddies and anticyclonic eddies is quite different. Yan et al. [17] suggested that the energy conversion terms play opposite roles during the evolution of cyclonic and anticyclonic eddies. The difference in Figure 5c implies different energy transfer during the interactions between the Kuroshio and the eddies.

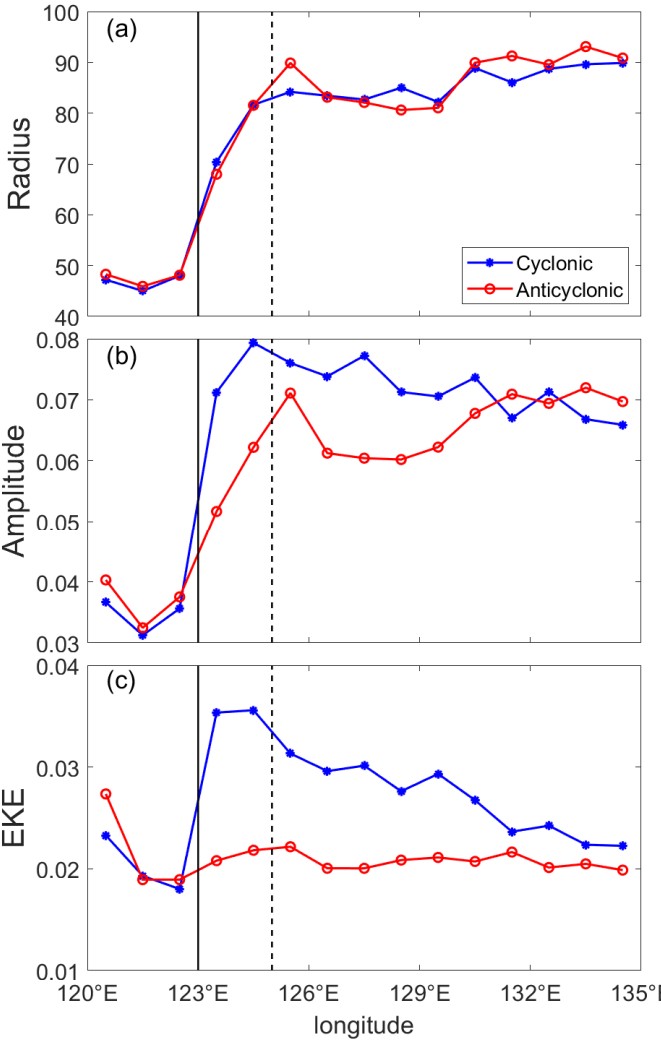

**Figure 5.** Zonal distribution of (**a**) eddy size (km), (**b**) amplitude (m), and (**c**) the EKE ($m^2/s^2$) averaged over meridional bands. The width of the bands for averaging is 1° longitude. The blue curve in each panel represents the parameters of cyclonic eddies, whereas the red curve represents those of anticyclonic eddies. Black solid (dashed) lines are located at 123° E (125° E), and eddies west (east) of the solid (dashed) lines are regarded as the same sample data.

To evaluate the significance of the difference in the mean eddy characteristics (radius, amplitude, and EKE) between the Kuroshio region and the ocean interior, we applied the t-test on the two samples, the eddies in the Kuroshio (120° E to 123° E) and in the ocean interior (125° E to 135° E). The statistical mean values in the Kuroshio region and the ocean interior passed the *t*-test. Statistical test *p*-values are 0.0672 for anticyclonic eddy radius samples, $1.376 \times 10^{-25}$ for cyclonic eddy radius samples, and almost 0 for eddy amplitude samples and cyclonic eddy EKE samples. A small *p*-value suggests that the mean values are significantly different between the Kuroshio region and the ocean interior.

### 3.3. Comparison of Eddy Characteristics between Eddies near the Kuroshio and in the Pacific Interior

To analyze the temporal variability of the eddies throughout their lifespan, we compared the characteristics of the eddies that can propagate to the Kuroshio region and those that decay in the western Pacific interior. In both cases, only eddies with lifespans longer than 6 weeks were chosen for the comparison. Eddies were considered to reach the Kuroshio if the edge of an eddy intersects with 123° E during its lifespan (Figure 1). We normalized each eddy's age by its lifespan. Similarly, the temporal evolution of the three parameters above was analyzed by normalizing such parameters by subtracting the minimum lifespan, then dividing by the maximum minus the minimum (their range).

Figure 6 shows the characteristics of eddies according to whether they reach the Kuroshio region or not. We treated the eddy characteristics in the western Pacific interior (WPI) as the basic states of the eddies. To analyze eddies' characteristic average temporal throughout the eddy's lifespans, only eddy lifespans more than seven weeks were included. In total, 737 cyclonic and 638 anticyclonic eddies were identified as the WPI eddies between 1993 and 2017. A normalized time coordinate was then defined with the lifespan of each eddy divided into timesteps of 1/40 of the lifespan. The radius, amplitude, and EKE of each eddy across its lifespan were then mapped onto this time coordinate using cubic spline interpolation. To show the statistical significance, the normalized variations of the radius, amplitude, and EKE for all the eddy tracks, as well as their mean curves are shown in Figure S1 (see the Supplementary Materials). For each eddy category, all associated eddies were composited onto a time evolution curve based on the normalized eddy lifespans. The characteristics of the WPI eddies were consistent with the results of Liu et al. [5] with near symmetry along their lifespan (red solid curves in Figure 6). It can be seen clearly that the eddy size, EKE, and eddy amplitude go through three stages. In the first 0–0.3 of an eddy's lifespan (the youthful stage), the eddy size, EKE, and amplitude increase and then remain stable for the next 0.3–0.7 of the lifespan (the mature stage). In the last 0.7–1.0 of the eddy lifespan, the parameters decrease considerably, with the same amplitude and timing of the rise and fall. Cyclonic eddies show a similar trend to anticyclonic eddies.

As a comparison, we selected the eddies that could arrive at the Kuroshio region (KC) and investigated the difference from the WPI eddies. A total of 66 cyclonic and 51 anticyclonic eddies were identified as the KC eddies (blue solid curves in Figure 6). In general, both the cyclonic and anticyclonic KC eddies showed obvious asymmetry along their lifespans. That is, the values of the radius, amplitude, and EKE increased with time, similarly to the WPI eddies, but decreased faster, and eddies persisted for longer with a smaller size, amplitude, and EKE than is possible in the interior. To better resolve the tendency of the characteristics of eddies during their life span, we calculated the absolute value of the time derivative of each variable for the eddies. For the first stage of the eddy lifespan, the growing speeds of both the KC eddies and WPI eddies were quite close (blue and red dashed curves in Figure 6). However, the decaying speeds of the size and amplitude of the KC eddies were higher than those of the WPI eddies in the last stage of the eddy lifespan. The difference of the decaying speed of the EKE for the WPI eddies was also larger than that of the WPI eddies, but not significant compared with the other two variables. The variation of the eddies suggests the influence of the main current of the Kuroshio. For cyclonic eddies, the radius, amplitude, and EKE reached their maximum near

0.45 of the eddy's lifespan. For anticyclonic eddies, the maximum was at 0.35. There were also fluctuations as they increased. The duration of the decline was longer for anticyclonic eddies than for cyclonic eddies. The difference between the cyclonic and anticyclonic eddies implies different mechanisms for the effect of the Kuroshio main current on cyclonic and anticyclonic eddies.

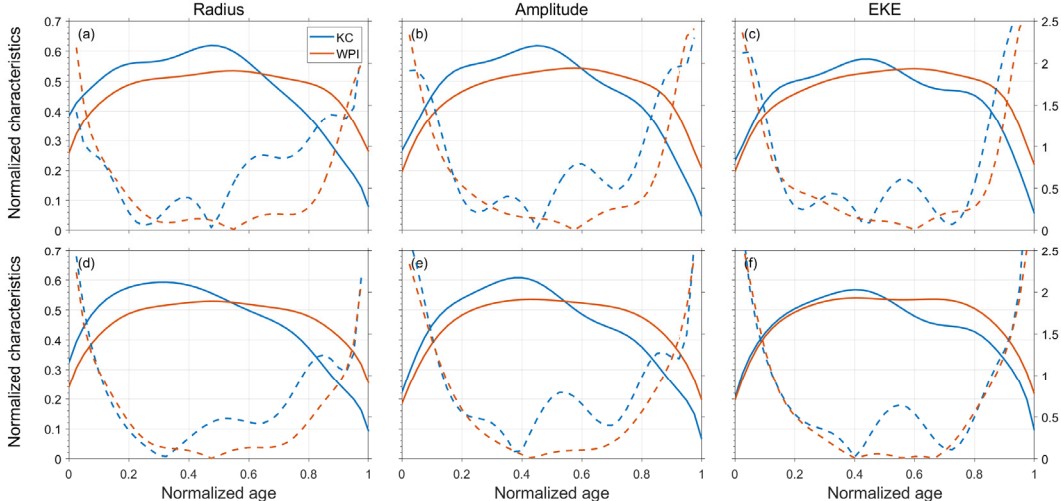

**Figure 6.** Time evolution of mean eddy characteristic parameters (solid curves, normalized by their range) and the absolute values of their time derivative (dashed curves): (**a**,**d**) radius, (**b**,**e**) amplitude, and the EKE (**c**,**f**). The blue curve in each panel represents the characteristics of eddies that reach the Kuroshio region (KC), whereas the red curve represents those in the western Pacific interior (WPI). Upper panels show cyclonic eddies, and lower panels show anticyclonic eddies.

## 4. Three-Dimensional Structure of Eddies Influenced by the Kuroshio

The analysis in Section 3 using the satellite altimeter data revealed that the characteristics of eddies are modified as they approach the Kuroshio region. As the structure of the eddies in the subsurface layer may also be influenced by the Kuroshio and might be different from their surface features, it is necessary to examine the vertical structure of the eddies in the vicinity of the Kuroshio. In the geostrophic balance assumption, the anomalies of the sea surface height in the eddies should be consistent with the variation of the density profiles in the upper ocean. Based on the linearized state equation of seawater, 90% of the density anomalies are contributed by the temperature anomalies in the eddies [35]. Therefore, we analyzed the three-dimensional OFES temperature data using the eddy detection method used in previous sections and composited the structures of eddies that encounter the Kuroshio (as defined in Section 3.3). In total, 82 eddy trajectories were collected from 1993 to 2017, including 48 cyclonic eddies and 34 anticyclonic eddies (Figure 7). We divided the region into four rectangular bands, 121°–123° E, 123°–125° E, 125°–127° E, and 127°–129° E, with meridional boundaries at 15° and 27° N (shown as brown boxes in Figure 7). We calculated the temperature anomalies of the eddies with their centers located in each band to show the variation of the eddies in different stages during their interaction with the Kuroshio. For the eddies in each band, we normalized the lateral dimensions of the eddies by the distance from the center to their edges. The temperature anomaly in each eddy was obtained by subtracting the climatological mean temperature at the eddies' location. As a consequence, the composited three-dimensional temperature anomalies in the eddies can be derived by interpolating the composited data onto a normalized high-resolution grid at each vertical level.

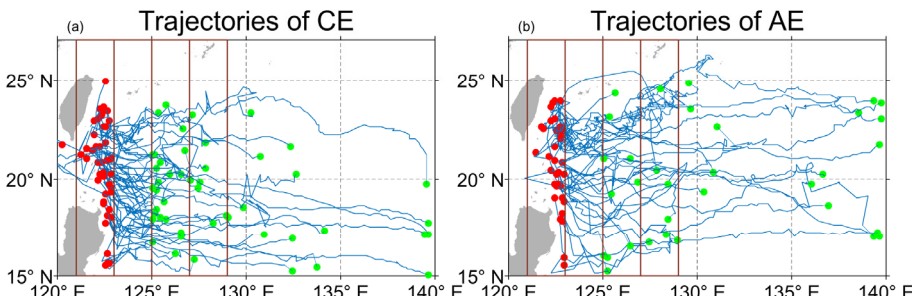

**Figure 7.** Trajectories of the detected (**a**) cyclonic and (**b**) anticyclonic eddies from the OFES dataset. Green (red) dots represent the locations of eddy generation (termination). Brown boxes represent the four longitudinal bands which are used for three-dimensional eddy composition.

In the ocean interior, the structure of the temperature anomalies of the eddies is steady (Figure 8b–d,f–h). However, the temperature anomalies in both cyclonic and anticyclonic eddies are clearly modified as the eddies approach the Kuroshio (Figure 8a,e). The maximum temperature anomalies in the eddies are at ~300 m (Figure 8i–l), which is the depth of the main thermocline, whereas the anomalies in the surface layer are not so obvious for both the cyclonic and anticyclonic eddies. The horizontal extent of the temperature anomalies (defined by temperature anomalies of 0.6 °C) increases from east to west, with the maximum depth reducing. In the 121°–123° E band range, eddies intrude the Kuroshio and the structures of the eddies spread out. If we assume the temperature anomalies induced by the eddies are an additional variation to the climatological state of the ocean, as the climatological mean state is subtracted, the variation of the temperature anomalies in the Kuroshio region should be induced by the interactions between eddies and the Kuroshio. The temperature anomalies are smaller in the 121°–123° E band compared with those in other bands, which suggests the size and amplitude of the eddies decrease and the eddies tend to decay in the Kuroshio region. This three-dimensional structures of the eddies in the Kuroshio region are consistent with the surface structure in Section 3.

To obtain a direct insight into the vertical structure of eddies, we computed three-dimensional temperature anomalies within the composite eddies. The result showed that fusiform eddies are turned into conical eddies and the maximum depth of both cyclonic eddies and cyclonic eddies becomes shallower and shallower. The maximum depth of the temperature anomalies decreases from ~300 m to ~200 m.

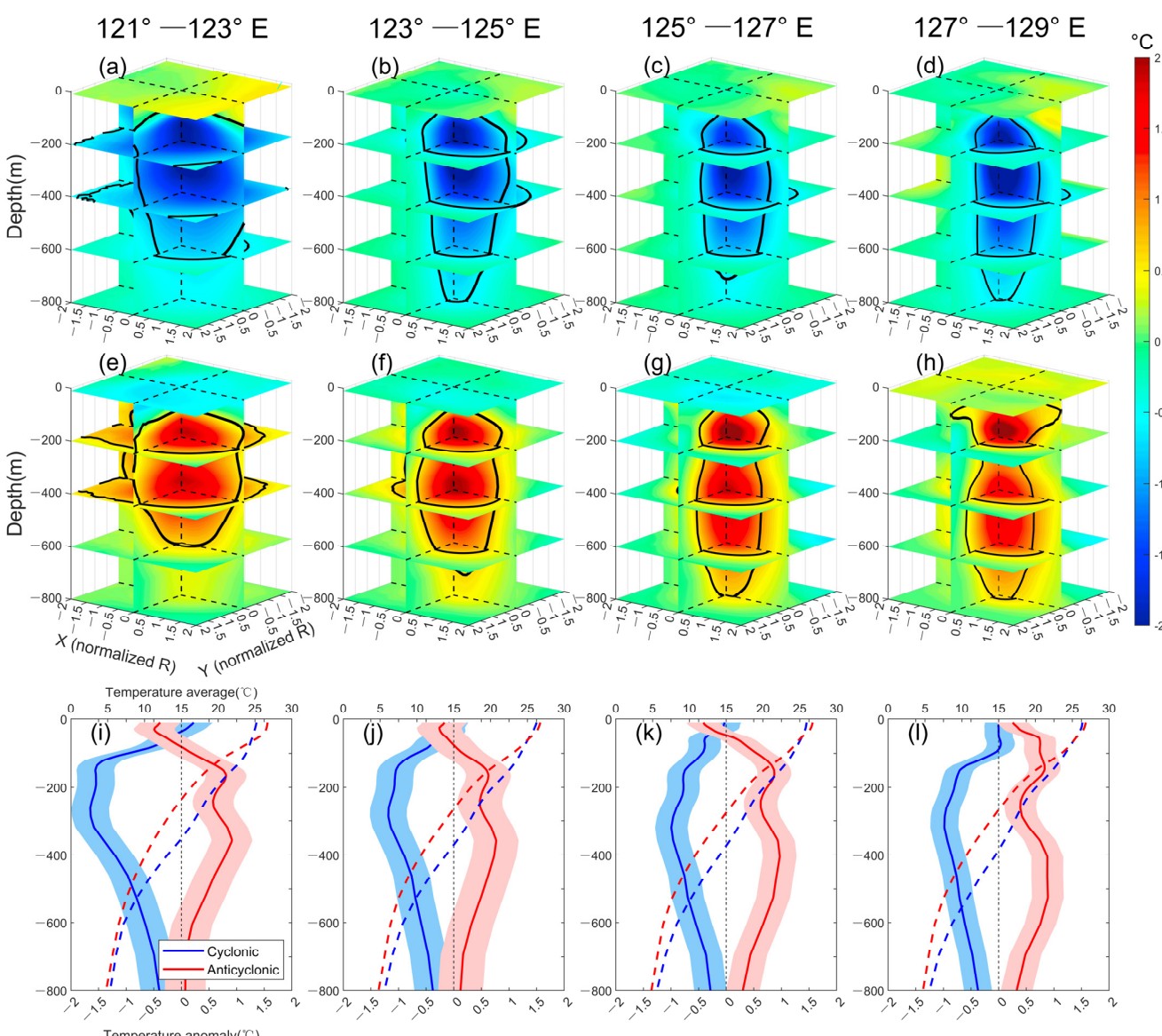

**Figure 8.** Temperature anomalies of composited cyclonic (**a–d**) and anticyclonic (**e–h**) eddies within different longitudinal bands. Black curves represent the boundaries of the eddy structure with 0.6° C temperature anomalies. X (Y) represents the normalized distance (evaluated by the eddy radius) in the zonal (meridional) direction. The vertical profiles of the averaged temperature (dashed curves) and its anomalies (solid curves) in each band are also shown (**i–l**). The shadows in (**i–l**) are standard deviations. Blue and red curves represent the cyclonic and anticyclonic eddies, respectively.

## 5. Conclusions

Using 25 years of satellite altimeter observational data and the OFES model data, we analyzed the variations of mesoscale eddies in the northwestern Pacific Ocean as they approach the Kuroshio region. Eddy detection based on vector geometry was applied to the SSHA-derived geostrophic currents to identify and track eddies. An eddy dataset was retrieved, including the eddy size, amplitude, EKE, and lifespan. More than 60% of the eddies have a radius of 40–60 km, and less than 17% have a radius larger than 100 km. The number of eddies increases toward the western boundary of the Pacific and reaches a maximum in the area between the south of Taiwan and north of Luzon Island. Anticyclonic eddies have a larger radius and smaller amplitude than cyclonic eddies. The eddy radius is large in the east and at a minimum in the Kuroshio, which is consistent with the spatial distributions of the EKE and eddy amplitude. The longitude distribution map was obtained

by averaging the spatial distribution map over latitude. Eddy size and amplitude are stable from 135° E to 125° E and decrease rapidly from 125° E to 121.5° E, which suggests that the eddies tend to decay as they interact with the main current of the Kuroshio. The variation of the EKE along the zonal direction is not significant; this may be because eddies obtain kinetic energy when they interact with the Kuroshio before they decay.

We also compared the characteristics of the eddies reaching the Kuroshio region with those in the Pacific interior. The size, EKE, and amplitude of the eddies that reach the Kuroshio region decrease sharply over the last 55% of their lifespan (0.45–1.0 of the eddy lifespan) under the interaction between the Kuroshio and the eddies. The duration of the decline for anticyclonic eddies is longer than for cyclonic eddies because the Kuroshio has different effects on the decay of cyclonic and anticyclonic eddies. However, the characteristics of the eddies in the internal region are symmetric with respect to their lifespan.

Using the 3D eddy-resolving OFES model data, the 3D structure of eddies was examined to study how eddies are influenced by the Kuroshio. In total, 48 cyclonic and 34 anticyclonic eddy trajectories from January 1993 to December 2017 were used to construct the composite 3D structures of eddies. The maximum depth of cyclonic eddies can exceed 700 m, and anticyclonic eddies are deeper than cyclonic eddies. When eddies are influenced by the Kuroshio, their maximum temperature anomalies become larger and shallower.

**Supplementary Materials:** The following Supporting Information can be downloaded at: https://www.mdpi.com/article/10.3390/jmse10121975/s1, Figure S1: Time evolution of mean eddy characteristic parameters.

**Author Contributions:** Conceptualization, D.C. and X.L.; methodology, Y.S., X.L. and T.L.; data curation, Y.S. and X.L.; software, Y.S., X.L. and T.L.; formal analysis, Y.S. and X.L.; investigation, X.L. and T.L.; writing-original draft preparation, Y.S.; writing-review and editing, X.L. and T.L.; supervision, D.C., X.L. and T.L.; funding acquisition, D.C. and X.L. All authors have read and agreed to the published version of the manuscript.

**Funding:** This work was supported by the Natural Science Foundation of China (41730535, 42176009), the Scientific Research Fund of the Second Institute of Oceanography, Ministry of Natural Resources under Contract No. JB2101, the Zhejiang Provincial Natural Science Foundation of China (LY21D060001), the Guangdong Basic and Applied Basic Research Foundation (2021B1515120080), and the Natural Science Foundation of China (42106008).

**Data Availability Statement:** The Satellite Altimeter data were obtained from AVISO web site. The model data were obtained from the Ocean general circulation model for the Earth Simulator (OFES) model data. The data are available at http://marine.copernicus.eu accessed on 28 September 2022 and http://apdrc.soest.hawaii.edu accessed on 28 September 2022.

**Acknowledgments:** We thank four anonymous reviewers for their helpful comments. We thank Tao Lian at SIO, MNR for his valuable advice on the statistical analysis.

**Conflicts of Interest:** The authors declare no conflict of interest.

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
