# Peer review of "Characteristics of Mesoscale Eddies in the Vicinity of the Kuroshio: Statistics from Satellite Altimeter Observations and OFES Model Data"

_jmse, doi:10.3390/jmse10121975_

Round 1

Reviewer 1 Report

A review of "Characteristics of Mesoscale Eddies in the Vicinity of the Kuroshio: Statistics from Satellite Altimeter Observations and OFES Reanalysis Data"

This paper describes the behavior of ocean eddies near the Kuroshio. The authors argue that the Kuroshio can exert significant impact on ocean eddies, including their radius and amplitude. This paper also describes the 3D structures of ocean eddies by using numerical models.

This paper is well written. But I have some comments on the main results of the paper. Thus, I would like to suggest major revision.

(A) Throughout the study, the authors need to show statistical significance. For example, the authors should show error bars in Figure 5 and Figure 6. In Figure 8, the authors need to discuss whether the modification is large or small from the viewpoint of statistical significance.

(B) Figure 5 and related discission:

Please describe the characteristics of the eddies more carefully and more in detail. I agree with the authors that, overall, the eddy size and amplitude decrease to the west of 125E (Line 194). However, they actually increase from 121E to 120E, which is inconsistent with the authors discussion.

More importantly, although the authors mention that EKE increases (line 198) around the Kuroshio, EKE actually decreases from 125E to 122E, and then increases from 122E to 120E (Fig. 5c). Please describe the figures more appropriately and give more reasonable interpretation.

(C)Figure 6 and related discussion:

The authors argue that radius, amplitude and EKE decrease rapidly near the Kuroshio in both cyclonic and anticyclonic eddies (line 237-240).  However, it is hard to identify the difference from the figures. It looks to me that the radius, amplitude and EKE of the eddies that reach to the Kuroshio starts decreasing in the earlier stage of their lifespan, rather than mora rapidly decreasing, compared with the interior eddies. Thus, could the authors show tendency (i.e., time derivative the quantities) and compare them quantitatively.

(D)Figure8 and related discission

I agree with the authors that the temperature anomalies of the eddies are modified toward the western part of the domain. However, these results are insufficient as evidence to suggest that the Kuroshio can affect the vertical structure of these eddies. The modification can be just a typical lifecycle of the eddies as they move westward. Why do the authors compare the 3D structures of the eddies near the Kuroshio with the interior eddies, as conducted in Fig. 6.

(E) lines 273-234

Please describe specifically. What aspect of the three-dimensional structures are "consistent" with the surface structure?

Author Response

Response to Reviewer 1 Comments

This paper describes the behavior of ocean eddies near the Kuroshio. The authors argue that the Kuroshio can exert significant impact on ocean eddies, including their radius and amplitude. This paper also describes the 3D structures of ocean eddies by using numerical models.

This paper is well written. But I have some comments on the main results of the paper. Thus, I would like to suggest major revision.

Response: We thank the reviewer for the insightful and constructive comments. We have carefully revised the manuscript according to the reviewer’s comments. Our responses are marked as red text.

Point 1: (A): Throughout the study, the authors need to show statistical significance. For example, the authors should show error bars in Figure 5 and Figure 6. In Figure 8, the authors need to discuss whether the modification is large or small from the viewpoint of statistical significance.

Response 1: Thanks for the valuable comment. The statistical significance is of great importance to evaluate the reliability of this study.

First, since the eddy size and the amplitude identified by the eddy detection algorithm covers a very wide range, it might be hard to calculate the errors in Figure 5. Instead, to evaluate the significance of the difference in the mean eddy characteristics (radius, amplitude and EKE) between the Kuroshio region and the ocean interior, we applied the t-test on the two samples, eddies in the Kuroshio (120°E to 123°E) and in the ocean interior (125°E to 135°E). The statistical mean values in the Kuroshio region and the ocean interior have passed the t-test, which suggested that the mean values are significantly different between the Kuroshio region and the ocean interior.

The characteristics values in Figure 6 were normalized by their range of their values. We used to choose the eddies with all the lifetime to get the composite parameters. However, following the reviewer’s suggestion, we found that the standard deviation might be too larger. In the revised Figure 6, we now only consider the eddies longer than 7 weeks for both KC eddies and WPI eddies for the composition. Furthermore, Since the two curves were overlapped in each panel in Figure 6, it might be messy if we add the error bars on the curves directly. Therefore, we added a figure in the supplementary (Figure S1) showing the mean values in the eddy lifespans with their standard deviations of the KC eddies and those of the WPI eddies, separately.

Finally, for Figure 8, we added the average temperature anomalies at each layer with their standard deviations in panels Figure 8(i-l).

Point 2 (B): Figure 5 and related discission:

Please describe the characteristics of the eddies more carefully and more in detail. I agree with the authors that, overall, the eddy size and amplitude decrease to the west of 125E (Line 194). However, they actually increase from 121E to 120E, which is inconsistent with the authors discussion.

More importantly, although the authors mention that EKE increases (line 198) around the Kuroshio, EKE actually decreases from 125E to 122E, and then increases from 122E to 120E (Fig. 5c). Please describe the figures more appropriately and give more reasonable interpretation.

Response 2: Previous studies suggested that the eddies east of the Luzon Strait could affect the looping path of the Kuroshio and generate eddies on the west part of the strait (Yang et al., 2020; Long et al., 2021). It was also reported that the EKE increases around the Kuroshio because both cyclonic and anticyclonic eddies can obtain kinetic energy from the Kuroshio during the interaction (Yan et al, 2022). We added some detailed descriptions on the variation of the eddy size, amplitude and EKE near 120°E-121°E in the Luzon Strait. Please see Line 208-227 for details. 

Point 3 (C) Figure 6 and related discussion:

The authors argue that radius, amplitude and EKE decrease rapidly near the Kuroshio in both cyclonic and anticyclonic eddies (line 237-240).  However, it is hard to identify the difference from the figures. It looks to me that the radius, amplitude and EKE of the eddies that reach to the Kuroshio starts decreasing in the earlier stage of their lifespan, rather than more rapidly decreasing, compared with the interior eddies. Thus, could the authors show tendency (i.e., time derivative the quantities) and compare them quantitatively.

Response 3: In order to evaluate the decreasing tendency of the eddies during their lifespan, we added the time derivatives absolute value of each variable in Figure 6. For the eddies in the Kuroshio region, the tendency of their radius and amplitude increases markedly in the last half of their lifespans. We also addressed these results in Line 289-298.

It is also worth noting that, the lifespan of the eddies was normalized by their lifetime in Figure 6. The reason why the radius, amplitude and EKE decrease at 0.4~0.5 of the lifespan might be related to the region we defined as the Kuroshio region. Since we defined an eddy that intersects with 123°E during its lifespan as the eddy in the Kuroshio region, this condition might turn out that most of the KC eddies are influenced by the Kuroshio during the middle stage of their lifespans, rather than in the decaying stage. Nevertheless, the difference in the variations of the eddy characteristics between KC eddies and the WPI eddies in Figure 6 suggests the influence of the Kuroshio on the eddies.

Point 4 (D): Figure8 and related discission

I agree with the authors that the temperature anomalies of the eddies are modified toward the western part of the domain. However, these results are insufficient as evidence to suggest that the Kuroshio can affect the vertical structure of these eddies. The modification can be just a typical lifecycle of the eddies as they move westward. Why do the authors compare the 3D structures of the eddies near the Kuroshio with the interior eddies, as conducted in Fig. 6.

Response 4: Since we divided the western Pacific region into several longitudinal bands and composited the eddies according to their positions, the eddies in each specific band should be in all possible lifespans (youthful or decaying). Therefore, the variation of the eddy characteristics in different bands should have excluded the influence of the lifespan after the composition in the band. The temperature structure in difference bands should mainly suggested the spatial variations of the eddies. Therefore, the different structures of the temperature anomalies between Figure 8a,d and Figure 8(b,c/e,f) should be mainly induced by the Kuroshio current. Furthermore, following the reviewer’s suggestion, we added another band in the east of the Pacific region (127°E to 129°E) in Figure 8 to show that the variation of the temperature structure in the ocean interior is quite smaller compared with that in the Kuroshio region. The results are consistent with that in Figure 6. We also addressed this in the text, Line 319; 328-329

Point 5 (E): lines 273-234

Please describe specifically. What aspect of the three-dimensional structures are "consistent" with the surface structure?

Response 5: In the 121°–123°E band range, eddies intrude the Kuroshio and the structures of the eddies spread out. The 3D structure of the temperature shows that the anomalies in the subsurface (mainly near the thermocline) are weaker in the Kuroshio region, and the strength of the eddies should be weaker compared that in the ocean interior. We could infer that the size and amplitude of the eddies in the region 121E-123E should be smaller. Therefore, we stated that the three-dimensional structures of the eddies in the Kuroshio region are consistent with the surface structure in Section 3. We addressed the detailed description in Line 340-343.

References:

Long, Y., Zhu, X.-H., Guo, X., Ji, F., & Li, Z. (2021). Variations of the Kuroshio in the Luzon Strait revealed by EOF analysis of repeated XBT data and sea level anomalies. Journal of Geophysical Research: Oceans, 126, e2020JC016849.

Yang, Q., Liu, H., Lin, P. et al. Kuroshio intrusion in the Luzon Strait in an eddy-resolving ocean model and air-sea coupled model. Acta Oceanol. Sin. 39, 52–68 (2020).

Yan, X.; Kang, D.; Pang, C.; Zhang, L.; Liu, H. Energetics Analysis of the Eddy–Kuroshio Interaction East of Taiwan. Journal Of Physical Oceanography 2022, 52(4), 647–664. https://journals.ametsoc.org/view/journals/phoc/52/4/JPO-D-21-0198.1.xml

Reviewer 2 Report

This is an interesting study that could make a contribution to mesoscale eddies research. 

I would support the publication as it is. Here are my minor corrections for specific line numbers. I hope the authors will find them useful.

79 horizontal resolution -> a horizontal resolution

281 normlized -> normalized

Author Response

This is an interesting study that could make a contribution to mesoscale eddies research. 

I would support the publication as it is. Here are my minor corrections for specific line numbers. I hope the authors will find them useful.

We thank the reviewer for the comments on our study. We revised the minor corrections as the reviewer suggested.

Point 1 : 79 horizontal resolution -> a horizontal resolution

281 normlized -> normalized

Response 1: Thanks for the correction. We revised the typo. Please see Line 80; Line 358.

Reviewer 3 Report

Please refer to the uploaded document 

Author Response

Thanks for the reviewer's effort on our manuscript. However, we could not find the attached material from the system.

Reviewer 4 Report

The manuscript presents a mostly well written account detailing various statistics of eddies in the Kuroshio western boundary layer region.  Eddy statistics include EKE, sea surface height, radius and number and were derived from the AVISO and OFES data sets. Despite the results being clearly presented, I feel that the paper is not ready for publication. Essentially, more thought needs to go into the choice of the statistics presented and how these are interpreted. Reading the paper in its current form, one arrives at the end with the feeling that not much has been learned. Specific comments follow:

1:  Fig 3; results for the cyclones and anticyclones look very similar. Are the differences statistically significant ?  I suspect the answer is no, in which case this should be stated explicitly. Related to this, various numbers are reported to high precision (e.g., line 150 "23.89% and 23.48%").  Surely these differences are also statistically insignificant and thus they should not be emphasized.

2: Fig 4: This is one of my main critical comments.  Clearly the results in the two left figure panels are skewed by a single pixel near the top left corner. Looking at the other panels, one suspects that the large number of eddies (of both signs) detected at this location are spurious. At any rate, they appear to be uninteresting (small, weak amplitude, etc).  This suggests that some kind of fine tuning of the eddy detection algorithm is needed. 

3:  I didn't see where the methodology for assessing EKE was discussed. 

4: Fig 5. I think this shows results for all eddies in the various latitude bands. As such, it is not clear whether, say, the reduction in radius as an eddy approaches the boundary is due to interaction with the boundary current or simply due to only the smaller eddies making it that far westward. Also, the distinction between behaviours of cyclones and anticyclones in the lower panel seems interesting and should be discussed in more depth.

5: I didn't see the point to Fig 6.

6: Section 4.  A few comments here. i) I assume temperature is a good proxy for density in this part of the ocean, but this is never stated (e.g., why are we looking at temperature instead of density perturbations?).  ii) The temperature structure in Fig 8 has sub-surface max/min (as mentioned in the text).  Why is this? Are these mode 2 eddies?  Basically, give some more information as to the basic structure of the eddies that are being discussed. iii) The anomalies are defined relative to a time mean. Given this, even if an eddy doesn't change its structure at all as it encounters the current, its anomaly will change (simply because the mean with respect to which it is defined has changed).  This should be discussed (i.e., more should be done to convince the reader that the eddies are changing due to interactions with the current as opposed to this being a simple consequence of how the eddies were defined). 

There are a number of minor stylistic things to be dealt with. The first would be to remove Title from the title.  Basically, a thorough proofreading is needed (but the writing is mostly well done).   

Author Response

Response to Reviewer 4 Comments

The manuscript presents a mostly well written account detailing various statistics of eddies in the Kuroshio western boundary layer region.  Eddy statistics include EKE, sea surface height, radius and number and were derived from the AVISO and OFES data sets. Despite the results being clearly presented, I feel that the paper is not ready for publication. Essentially, more thought needs to go into the choice of the statistics presented and how these are interpreted. Reading the paper in its current form, one arrives at the end with the feeling that not much has been learned. Specific comments follow.

Response: We thank the reviewer for the insightful and constructive comments and suggestions. We have carefully revised the manuscript according to the reviewer’s comments. We added more discussion on the statistical methods and the significance of the results according all the reviews’ suggestions. Furthermore, our study was aimed to give the readers information that the western boundary is the destination of the mesoscale eddies, and the eddies are influence by the Kuroshio as they are approaching the west boundary region. The results in this study might give us inspirations that the processes during interactions between the Kuroshio and eddies, such as energy transfer and potential vorticity budget, could be very important for each other. Our responses are marked as red text.

Point 1: Fig 3; results for the cyclones and anticyclones look very similar. Are the differences statistically significant?  I suspect the answer is no, in which case this should be stated explicitly. Related to this, various numbers are reported to high precision (e.g., line 150 "23.89% and 23.48%").  Surely these differences are also statistically insignificant and thus they should not be emphasized.

Response 1: Thanks for the comment on the statement of the statistical results. We admit that the differences are statistically insignificant. We added more description on this fact in the text, Line 144-153. We also revised the statement of the statistic values between AEs and CEs in the text following to the reviewer’s suggestion, Line 145-161.

Point 2: Fig 4: This is one of my main critical comments.  Clearly the results in the two left figure panels are skewed by a single pixel near the top left corner. Looking at the other panels, one suspects that the large number of eddies (of both signs) detected at this location are spurious. At any rate, they appear to be uninteresting (small, weak amplitude, etc).  This suggests that some kind of fine tuning of the eddy detection algorithm is needed. 

Response 2: Thanks for the valuable comment. Indeed, we did not consider carefully on the abnormal values in Figure 4 which should be caused by the inaccuracy of the eddy detection algorithm near the coastal region due to the shallower water depth. Therefore, we revised Figure 4 and masked the grid points where the water depth is less than 300 meters to avoid the errors in the coastal region. Furthermore, we revised Figure 5 which is related to the result of Figure 4. We found that there is no significant effect on the overall statistic results.

Point 3:  I didn't see where the methodology for assessing EKE was discussed. 

Response 3: We added the calculation of EKE in the methodology section. Line 125-129.

Point 4: Fig 5. I think this shows results for all eddies in the various latitude bands. As such, it is not clear whether, say, the reduction in radius as an eddy approaches the boundary is due to interaction with the boundary current or simply due to only the smaller eddies making it that far westward. Also, the distinction between behaviors of cyclones and anticyclones in the lower panel seems interesting and should be discussed in more depth.

Response 4: As suggested by previous study (e.g. Chelton, et al., 2011; Liu et al., 2012), the mesoscale eddies propagate westward in the upper ocean, with most of their lifetimes as several weeks. Particularly in the subtropical zone of the North Pacific Ocean, there is no significant difference if we compare the eddy size zonally in the ocean interior (See the Figure 4 and Fig.6 of Liu et al., 2012). Further, the eddies are generated all over the ocean interior including the region near the western boundary region (See Fig.9 of Liu et al., 2012). Therefore, the deduction of the eddy size and amplitude should be induced by their interaction with the Kuroshio. We may infer that, suppose the Kuroshio did not exist, the eddies could propagate westward and cross the Luzon Strait. As for the distinction between behaviors of cyclonic EKE and anticyclonic EKE, the energy transfer during the interaction between eddies and Kuroshio was investigated by previous studies (Yan et al., 2022). The dynamics was different between the cyclonic eddies and anticyclonic eddies. We added more discussion about the difference between the EKE of CEs and AES, Line 248-255. 

Point 5: I didn't see the point to Fig 6.

Response 5: As suggested in the text, the variations of the radius, amplitude and EKE of the eddies which could reach the Kuroshio region are different from those in the ocean interior. In the ocean interior, all the characteristics of the eddies are nearly symmetric along their normalized age (red solid curves). While in the Kuroshio region, the decaying tendency of these characteristics are larger than the growing stage (blue solid curves). To show the tendency in different stages clearly, for both the eddies in Kuroshio region and eddis in the Pacific interior, we added the time derivative absolute value about these characteristics (dashed curves in Figure 6). We found that the time derivative absolute value of these characteristics are larger in the last half of the lifespans. We also revised the description on Figure6 in the text, Line 289-298.

Point 6: Section 4.  A few comments here. i) I assume temperature is a good proxy for density in this part of the ocean, but this is never stated (e.g., why are we looking at temperature instead of density perturbations?).  ii) The temperature structure in Fig 8 has sub-surface max/min (as mentioned in the text).  Why is this? Are these mode 2 eddies?  Basically, give some more information as to the basic structure of the eddies that are being discussed. iii) The anomalies are defined relative to a time mean. Given this, even if an eddy doesn’t change its structure at all as it encounters the current, its anomaly will change (simply because the mean with respect to which it Is defined has changed).  This should be discussed (i.e., more should be done to convince the reader that the eddies are changing due to interactions with the current as opposed to this being a simple consequence of how the eddies were defined). 

Response 6:

i) Thanks for the suggestion. We added the statement explicitly that the temperature anomalies in the eddies are a good estimation for the density variation. Line 310-314.

ii) The subsurface maximum variability in the eddies could be induced by the vertical movement of thermocline, which is mainly caused by the circulation of the eddies (He et al., 2018). As the vertical variation of the temperature is maximum near the thermocline, the anomalies could be the most significant near the thermocline depth. We added the average temperature in the eddies in Figures 8 (i-l) to show the basic states of the eddies. We also added more discussion on that, Line 328-331.

iii) The temperature anomaly of one eddy is calculated by subtracting the climatological mean value at the eddy’s location. Our main consideration is that we assume the temperature anomalies is the additional process to the climatological state of the ocean. As in the Kuroshio region, suppose there is no interaction between the Kuroshio and the eddy, the temperature field itself should be the sum of the anomalies within the eddy and the thermocline tilting of the main circulation. Therefore, the spatial variation of the temperature anomalies with respect to the climatological mean should be reasonable evidence for the influence of the Kuroshio on the eddies. We added more discussion about this, Line 336-343.

Point 7: There are a number of minor stylistic things to be dealt with. The first would be to remove Title from the title.  Basically, a thorough proofreading is needed (but the writing is mostly well done).

Response 7: Sorry for the inconvenience. We checked the typos and minor errors in the whole text and made a thorough proofreading before resubmission. Thanks for the comment.

References:

Chelton, D.B.; Schlax, M.G.; Samelson, R.M. Global observations of nonlinear mesoscale eddies. Progress in Oceanography 2011, 91(2), 167–216. DOI: 10.1016/j.pocean.2011.01.002

Liu, Y.; Dong, C.; Guan, Y.; Chen, D.; McWilliams, J.; Nencioli, F. Eddy analysis in the subtropical zonal band of the North Pacific Ocean. Deep Sea Res Pt I 2012, 68, 54–67.

Round 2

Reviewer 1 Report

The authors have addressed my concerns  appropriately. Threfore, I would like to suggest acceptance.

Reviewer 4 Report

I think the manuscript is improved following the authors' consideration of reviewer comments from the previous round and now recommend acceptance.